# Measurement Report: Influence of particle density on secondary ice production by graupel and frozen drop collisions

Sudha Yadav[1], Lilly Metten[1], Pierre Grzegorczyk[2], Alexander Theis[3], Subir K. Mitra[3], and Miklós Szakáll[1]

[1]Institute for Atmospheric Physics, Johannes Gutenberg University, Mainz, Germany
[2]Laboratoire de Météorologie Physique (UMR6016)/UCA/CNRS, Aubière, France
[3]Particle Chemistry Department, Max Planck Institute for Chemistry, Mainz, Germany,

**Correspondence:** Miklós Szakáll (szakall@uni-mainz.de)

**Abstract.** Collision induced fragmentation of atmospheric ice particles is a crucially important, yet still understudies secondary ice production mechanism in clouds. We present a laboratory study dedicated to fragmentation due to graupel-graupel and frozen drop-frozen drop collisions and their role in augmenting ice particle concentration in clouds. For this, graupels of different sizes and densities were created utilizing dry growth condition in a cold chamber at -7 °C and -15 °C using a setup that simulates the natural rotation and tumbling motion of freely falling graupels. Ice spheres as proxies for frozen drops and ice pellets were generated by freezing purified water in 3D-printed spherical molds. We conducted collision experiments inside the cold chamber utilizing a fall tube that ensures central and repeatable collision of ice particles at different collision kinetic energies. The number of fragments generated in the collisions were analyzed following a theoretical framework as a function of the collision kinetic energy. The detection limit of our experiments was 25 to 30 $\mu$ m, thus fragments with sizes lower than that could not be detected. Our results revealed a strong dependency of the fragment number on the density of the colliding ice particles, which can be attributed to the particles' structure. The observed number of fragments varies between 1 and 20 and, thus, comparable or larger than those resulted in drop freezing experiments. The size of the fragments we detected was in the sub-mm range for graupels, and up to 3 mm for ice spheres. Another set of experiments, focusing on multiple collision of graupel revealed that the number of fragments generated decreases significantly and approaches zero when the particle undergoes more than three collisions in a row.

## 1  Introduction

Ice particles are formed in the atmosphere either by primary processes like homogeneous freezing or heterogeneous ice nucleation from the liquid or vapor phase, or by secondary ice processes (SIP), which is the generation of ice crystals during processes that involve already existing ice particles. There is a consent that SIP (also known as ice multiplication) is responsible for the observed discrepancy between the number of detected ice crystals and ice nucleating particles (INPs) in the

atmosphere at temperatures between $-5$ and $-15\,°C$ (Field et al., 2017; Ladino et al., 2017; Sullivan et al., 2017; Korolev et al., 2022). In a recent study, Korolev et al. (2022) provided in situ evidence of SIP occuring at $-27\,°C$ aloft. During an ice pellet storm, Lachapelle and Thériault (2022) attributed the observed discrepancy between INP and ice crystal number concentra-
tions to SIP. In spite of their crucial importance, the basic understanding of SIP is still lacking due to the scarcity of systematic laboratory studies (Korolev and Leisner, 2020). Currently, seven different mechanisms are distinguished as possible SIP processes. The most prominent for decades, has been the rime splintering (also known as Hallett-Mossop, hereinafter abbreviated as H-M process) in which ice splinters are produced upon accretion of $D < 13\,\mu m$ and $D > 24\,\mu m$ droplets by ice particles between $-3\,°C$ and $-8\,°C$ (e.g. Hallett and Mossop, 1974; Mossop, 1976, 1985; Choularton et al., 1978, 1980; Heymsfield
and Mossop, 1984). However, the relevance of this process in the atmosphere became debated recently. Seidel et al. (2024) conducted rime-splintering experiments under H-M conditions and detected no ice crystal formation. The second promising SIP process is the fragmentation during freezing (Lauber et al., 2018; Keinert et al., 2020). Lauber et al. (2018) distinguished between ice particle ejection and splitting during drop freezing. Although the number of the generated secondary ice particles is strongly drop-size dependent, it can exceed unity ($\sim 2.4$), especially bubble bursting might be very productive. Further SIP,
like ice fragmentation due to sublimation (Oraltay and Hallett, 1989), due to activation of INPs in transient supersaturation (Prbhakaran et al., 2020), due to thermal shock (Dye and Hobbs, 1968), and during break-up of freezing droplets on impact of ice particles (James et al., 2021) have been supported experimentally but remained uncharacterized yet.

In a recent study (Grzegorczyk et al. (2023); hereinafter G2023), we investigated the seventh prominent SIP, the fragmentation due to collision of ice particles. That study was in some sense a follow-up experiment of earlier investigations of Takahashi
et al. (1995) and Vardiman (1978). Motivated by the analysis of the dendritic growth zone by long-term radar observations of von Terzi et al. (2022), the collision of snowflake proxies were studied. For that, we used graupel particles with dendritic crystals grown on their surface, and natural-like snowflakes produced from dendritic crystals grown in an aquarium setup. Our G2023 study revealed that hundreds of ice fragments can be generated when particles with dendritic crystal structures are breaking apart upon collision. The fragment numbers were characterized in terms of the collision kinetic energy (CKE), which
is defined as

$$CKE = \frac{1}{2}\frac{m_1 m_2}{m_1 + m_2}(\Delta v)^2 \tag{1}$$

where $m_1$ and $m_2$ are the masses of the colliding particles, and $\Delta v$ is their fall velocity difference.

Phillips et al. (2017) (hereinafter P2017) introduced a full physical formulation of mechanical breakup of ice particles due to collision, which is based on an energy conservation principle. In this theoretical framework, the number of fragments $N$
generated by collision is calculated as

$$N = \alpha A \left( 1 - \exp\left( - \left[ \frac{C \cdot CKE}{\alpha A} \right]^{\gamma} \right) \right) \tag{2}$$

where $\alpha$ is the surface area of the smaller colliding particle; $A$ is the number of breakable asperities per unit area; while $C$ is the asperity-fragility coefficient, which is an inverse measure of the average work to break each branch or other asperity. Both $A$ and $C$ are dependent on the morphology of the more fragile particle in the collision, thus, on temperature, maximum

dimension, rime fraction, and particle type (i.e. graupel, snow, or hail). In G2023 and in its corrigendum we provided the parameters $A$, $C$, and $\gamma$ for graupels with dendrites on their surface, bare graupels, and snowflakes. In that study we presented indication for a dependency of the fragmentation on the particle structure, i.e. if the graupel particle had dendritic crystals on their surface or not.

The G2023 study also revealed that bare graupel particles, i.e. those which were produced by riming only, are able to produce fragments when they collide. Griggs and Choularton (1986) pointed out in their experimental study that the fragility of rime increases as the temperature is lowered. Moreover, they suggested that in a subsaturated environment, sublimation would further weaken the structure making the rimed graupel more fragile. Similarly, the morphology of ice particles, in terms of their rime fraction, was suggested to affect the number of fragments produced during collision in a recent study of Gautam et al. (2024).

In the present paper we aim to systematically investigate the effect of bare ice particle structure on the fragmentation during collision using low and moderate density graupels as well as frozen drops (representative for ice pellets). This study extends our earlier investigation G2023 with graupels having dendritic crystals grown on their surface. Furthermore, we investigate the effect of multiple collision of graupels on the ice multiplication mechanism.

## 2   Experimental

The experimental studies were performed inside the walk-in chamber of the Mainz vertical wind tunnel laboratory. Graupel and ice sphere generation and collisions were carried out at $-5\,°\text{C}$, $-7\,°\text{C}$ and $-15\,°\text{C}$. These temperatures correspond to regions where rimed particles are observed in mixed-phase clouds (Waitz et al., 2022). Furthermore riming can also occur at even lower temperatures, (e.g. down to $-20\,°\text{C}$; see in Tridon et al., 2022). The graupel particles generated at $-7\,°\text{C}$ and $-15\,°\text{C}$ (as in G2023) possess different morphological structures and, therefore, different densities. This is because supercooled droplets freeze faster at lower temperatures than at higher, leading to more air inclusion into the graupel structure, making its density lower (Enzmann et al., 2011).

### 2.1   Graupel generation

Rimed lump graupels were generated using the graupel generator GEORG (GEnerator instrument Of Rimed Graupel), which is described in G2023 in detail and originally introduced in Theis et al. (2022). The graupel generation is performed inside a 2-m-high flow tube having a cross-sectional area of 17 cm × 17 cm. The parameters used for generating the graupels in the present study are summarized in Table 1. A small epoxy sphere acts as an ice embryo which is exposed to a supercooled droplet stream, and the graupel grows by riming. Droplets with diameters of $23.4 \pm 8.8\ \mu$m are produced by an ultrasonic atomizer (US 2/58 Hz, Lechler GmbH, Germany) using a 30 L/min nitrogen flux, and injected to form a central droplet stream inside the flow tube. A blower is mounted at the top of the equipment and ensures a flow speed of $2.78 \pm 0.10$ m/s to simulate graupel free fall. The liquid water content (LWC) was $0.368 \pm 0.028$ g m$^{-3}$ which establishes dry growth conditions, and represents typical conditions in mixed-phase clouds. To minimize the effect of turbulence inside GEORG, which was very likely responsible for

**Table 1.** Experimental parameters for graupel generation in GEORG in the present study.

| Parameter | Value |
|---|---|
| air temperature | $-7\,°\mathrm{C}$ |
| RH (ice) | $92--100\%$ |
| air speed | $2.78 \pm 0.10\ \mathrm{m/s}$ |
| LWC | $0.368 \pm 0.028\ \mathrm{g m^{-3}}$ |
| mean droplet size | $23.4 \pm 8.8\ \mathrm{\mu m}$ |
| rotation, gyration frequency | 4 Hz |
| growth time | 10 min |

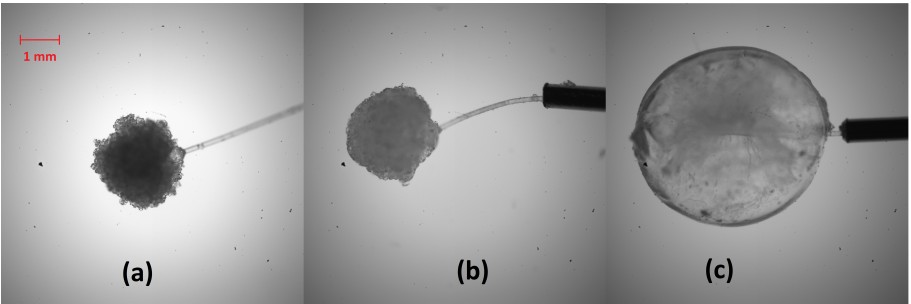

**Figure 1.** Microscope images of graupels with (a) a diameter of 2.23 mm and a density of 0.334 g cm$^{-3}$, grown at -15 °C and 0.723 g m$^{-3}$ LWC for 10 minutes in G2023; (b) a diameter of 2.4 mm and a density of 0.558 g cm$^{-3}$, grown at -7 °C and 0.368 g m$^{-3}$ LWC for 5 minutes in the present study; and (c) an ice sphere of 5 mm diameter and a density of 0.89 g cm$^{-3}$, generated at -15 °C. The red bar indicates 1 mm.

a relatively large scatter in graupel size and density during the preparation phase of our experiments, a honeycomb structure was placed at the bottom of GEORG, from where the air is sucked in. The setup can simulate the falling and tumbling motion of a graupel in natural clouds by a double gyration motor apparatus to which the graupel is attached. This apparatus consists of
one motor rotating around the vertical axis at 4 Hz, and a second one fixed at 45° angle with respect to the first one and rotating at the same rotational speed.

For the present experiments we generated graupels at $-7$ °C having an average diameter of 2.45 $\pm$ 0.28 mm and a density of 0.558 $\pm$ 0.02 g cm$^{-3}$ . In the experiments of G2023, graupels were 2.48 $\pm$ 0.17 mm, 2.12 $\pm$ 0.1 mm, and 4.0 $\pm$ 0.2 mm in diameter with densities of 0.21 $\pm$ 0.05 g cm$^{-3}$, 0.334 $\pm$ 0.062 g cm$^{-3}$ and 0.51 $\pm$ 0.07 g cm$^{-3}$. The low density of 0.21 g
95    cm$^{-3}$ was achieved by growing dendritic crystals on the surface of the graupel particle (see G2023 for details). Furthermore, the G2023 experiments were conducted at -15 °C. Images of graupels generated for the current measurements as well as in G2023 are depicted in Fig. 1a and b.

## 2.2 Generation of frozen drops

Ice spheres as proxies for frozen drops were generated by freezing Milli-Q water in a 3D-printed spherical mold inside a deep
freezer at -70 °C. Before the collision experiments, the mold was kept at the cold room temperature of either -5 °C or -15
°C for several minutes, then briefly held and warmed in both palms to facilitate its easy removal. The ice spheres used in the
collision experiments were 5 mm (shown in Fig. 1c.) and 7 mm in diameter, and had densities of $0.89 \pm 0.02$ g cm$^{-3}$, i.e.
corresponding to atmospheric small hail densities (Lachapelle and Thériault, 2022). The different experimental temperatures
enabled a systematic study of the effect of temperature and structural variations of ice on fragmentation.

## 3 Collision experiments and fragment analysis

The collision experiments were carried out using the same setup as in G2023 (see their Fig. 6) which consists of a "collision
tube" and a "fall tube". In the measurements, one particle is falling, while the other is kept stationary in the collision tube
using a thin plastic wire fitted into a cannula. The wire is flexible and possesses a small aerodynamic resistance, therefore, the
particle can freely move after collision. Hence, the collision energy is not consumed by the mounting and fed only into the
110 kinetic energies of the colliding particles and into fragmentation, which we could prove by high-speed video recordings. The
8-mm-thin fall tube guides the falling particle to the impact point and ensures a central collision between the two particles.

One aim of the present study was to derive a relationship between CKE and the number of fragments generated. Therefore,
collisions with different CKEs but same particle types and masses were carried out. For that, the falling particle was released
from different heights of 5 cm, 22 cm, and 80 cm. For each particle type and size, and for each fall height the particles' velocities
were determined from high speed (2000 frames-per-seconds) images recorded in a characterization set of measurements. The
masses and sizes of the graupel particles were also measured during the characterization measurements after producing several
particles in GEORG under predefined conditions of -7 °C temperature, 0.368 g cm$^{-3}$ LWC, and 10 min growth time. In case of
ice spheres, masses of several 5 mm and 7 mm diameter particles were measured and averaged. Densities were calculated from
particle mass and size. During the test experiments we observed that the epoxy inside a falling graupel modified the center of
120 mass of the particle. This resulted in a fall orientation in which the unrimed epoxy hit the fixed graupel. In order to avoid this
unnatural collision mode, we removed the epoxy from the falling graupel particle before the collision experiments. Therefore,
the density of the falling graupel was $0.46$ gcm$^{-3}$, i.e. less than that of the fixed one (see Table 2).

To investigate the effect of a graupel experiencing multiple collisions with other ice particles in the cloud, we conducted
experiments where a 2.45-mm graupel and a 5-mm ice particle were collided up to six times. The multiple collision experiments
were performed between graupel and ice sphere at the highest CKE, i.e. at a fall height of 80 cm, in order to account for the
maximum possible number of fragments that are produced. The number of collision repetitions was determined based on
the observation that after a certain number of collisions, no new fragments were observed (considering the detection limit of
approx. 25 $\mu$m), regardless of further collisions. For each subsequent collisions, a new frozen drop was used as the falling
particle. The fall tube ensured that the falling particles hit the fixed one approximately at the same position.

**Table 2.** Main characteristics of the collision experiments. GG: graupel-graupel collisions, II: ice sphere – ice sphere collisions, GI: graupel-ice sphere collisions, MC: multiple collisions. (Note, that ice spheres were used as proxies for frozen drops in the atmosphere.)

| Exp# | Collision type | Temperature (°C) | Fixed particle Size (mm) | Fixed particle Density (g/cm³) | Falling particle Size (mm) | Falling particle Density (g/cm³) | Fall height (cm) | CKE ($\mu$J) |
|---|---|---|---|---|---|---|---|---|
| 1 | II |  | 5.00 | 0.89 | 7.00 | 0.89 | 22 | 65.35 |
| 2 | II | -5 | 5.00 | 0.89 | 7.00 | 0.89 | 80 | 267.46 |
| 3 | II |  | 5.00 | 0.89 | 5.00 | 0.89 | 22 | 71.92 |
| 4 | II |  | 5.00 | 0.89 | 5.00 | 0.89 | 80 | 203.79 |
| 5 | II |  | 5.00 | 0.89 | 7.00 | 0.89 | 22 | 65.35 |
| 6 | II | -15 | 5.00 | 0.89 | 7.00 | 0.89 | 80 | 267.46 |
| 7 | II |  | 5.00 | 0.89 | 5.00 | 0.89 | 22 | 71.92 |
| 8 | II |  | 5.00 | 0.89 | 5.00 | 0.89 | 80 | 203.79 |
| 9 | GG |  | 2.45 | 0.558 | 2.45 | 0.46 | 5 | 0.95 |
| 10 | GG |  | 2.45 | 0.558 | 2.45 | 0.46 | 22 | 3.15 |
| 11 | GG |  | 2.45 | 0.558 | 2.45 | 0.46 | 80 | 7.70 |
| 12 | GI | -7 | 2.45 | 0.558 | 5.00 | 0.89 | 5 | 1.96 |
| 13 | GI |  | 2.45 | 0.558 | 5.00 | 0.89 | 22 | 15.37 |
| 14 | GI |  | 2.45 | 0.558 | 5.00 | 0.89 | 80 | 27.13 |
| 15 | MC |  | 2.45 | 0.558 | 5.00 | 0.89 | 80 | 27.13 |

In total, we conducted 15 series of experiments as listed in Table 2. Each series of experiments consisted of at least three individual collisions. The colliding particles were either ice spheres and ice spheres (II), graupels and graupels (GG), or graupels and ice spheres (GI). Particle sizes, densities, and the corresponding CKEs for each experimental series can be taken from Table 2. CKEs for graupels and spheres match the atmospheric values of these sizes when calculating the fall velocities using, e.g., the parameterization in Heymsfield et al. (2020), although they represent the highest end of the possible natural collision kinetic energies.

The ice fragments generated by collision were collected into a petri dish, which was filled with food grade mineral oil (density equal to $0.78\,\mathrm{g cm^{-3}}$ at room temperature) and placed at the bottom of the collision tube. The ice fragments were then analyzed under a microscope. In the case of multiple collisions, the petri dish was immediately removed after the collection of fragments during each collision and the images were analysed at the end of the experiment. For all collisions, total number of fragments, as well as fragment sizes were determined following the procedure described in G2023. We note here that the number of fragments with detectable sizes was low for all collision types, therefore, it was not possible to provide any fragment size distribution from this study. Instead, we will provide the minimum and maximum fragment sizes detected for each collisions. The size resolution of the applied optical setup was 3.8 $\mu$m-per-pixel. Nevertheless, we could not prevent the oil bath from being contaminated by dust particles (see Fig. 8b in G2023). This results in a detection limit for our analysis of 25 to 30 $\mu$m. Ice fragments larger than this size could be identified as ice particles with high reliability.

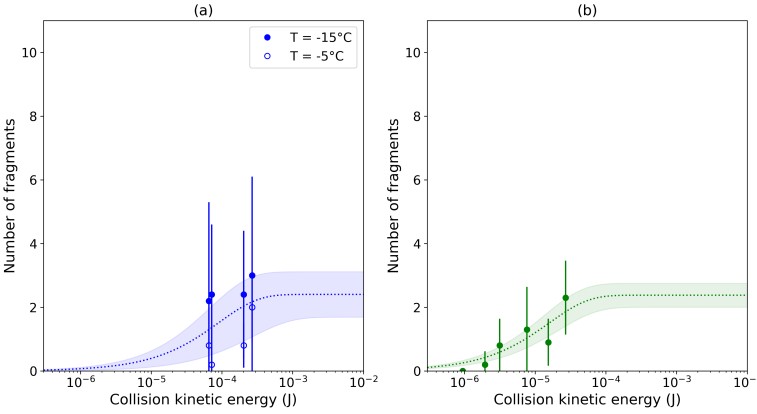

**Figure 2.** Number of fragments generated as a function of CKE during (a) ice sphere–ice sphere collisions at $-5\,°\text{C}$ and $-15\,°\text{C}$ (Exp# 1-8); and (b) during graupel–graupel and graupel–ice sphere collisions at $-7\,°\text{C}$ (Exp# 9-14). The dashed lines represent fits on data using Eq. 2, while the shaded areas are the $1\sigma$ standard deviation intervals of the fits. (Note, that ice spheres were used as proxies for frozen drops in the atmosphere.)

## 4    Results and discussion

### 4.1    Number of fragments after collision

Figure 2 shows the variation in the average number of fragments generated during the collisions as a function of the collision kinetic energy. The ice sphere – ice sphere collisions are presented in blue in Fig. 2a, whereas graupel–graupel and graupel–ice sphere collisions are shown in green in Fig. 2b. The average number of fragments released in each series of experiments are also provided in Table 3 together with their maximum and minimum values.

As expected, in both series of collisions the average number of fragments were observed to increase with increasing CKE. The data were fitted with the parameterization given in Eq. 2 and are represented by blue and green dashed lines in Fig. 2. The shaded areas in the two plots represent a $1\sigma$ standard deviation interval of the best fit curves. For fitting, the shape parameter $\gamma$ was held constant at $0.78$, which was also derived in G2023 for graupel–graupel with dendrites and graupel-snowflake collisions. Fitting the other parameters provided the values $A = 30610\,\text{m}^{-2}$ and $C = 26534.5\,\text{J}^{-1}$ for ice sphere – ice sphere, and $A = 126168\,\text{m}^{-2}$ and $C = 150597\,\text{J}^{-1}$ for graupel–graupel collisions.

Ice sphere–ice sphere collisions were conducted at two different temperatures, namely at -15 °C (shown with filled symbols in Fig. 1a) and at -5 °C (open symbols). There seems to be a temperature dependency of the number of fragments, and it is by trend higher at lower temperature in the investigated temperature range (see also Table 3). This temperature dependency is similar to that found by (Takahashi et al., 1995). Nevertheless, the difference between the number of fragments at these two temperatures is within the error of the experiments. Therefore, we combined the two dataset for further discussion.

**Table 3.** Average, minimum and maximum number of fragments, and minimum and maximum fragment sizes produced in each series of experiments. Collision type and temperature are also indicated.

| Exp# | Collision type | Temperature (°C) | Frag. number mean (min;max) | Frag. size (mm) min | max |
|---|---|---|---|---|---|
| 1 | II | | 0.8 (0;3) | 0.23 | 0.59 |
| 2 | II | -5 | 2 (0;3) | 0.46 | 2.05 |
| 3 | II | | 0.2 (0;1) | 0.52 | 0.52 |
| 4 | II | | 0.8 (0;2) | 0.33 | 0.76 |
| 5 | II | | 2.2 (0;8) | 0.09 | 1.00 |
| 6 | II | -15 | 3 (0;9) | 0.12 | 1.07 |
| 7 | II | | 2.4 (0;6) | 0.16 | 0.68 |
| 8 | II | | 2.4 (1;6) | 0.27 | 2.93 |
| 9 | GG | | 0 (0;0) | – | – |
| 10 | GG | | 0.8 (0;2) | 0.04 | 0.20 |
| 11 | GG | | 1.3 (0;4) | 0.02 | 0.24 |
| 12 | GI | -7 | 0.2 (0;1) | 0.09 | 0.15 |
| 13 | GI | | 0.9 (0;2) | 0.02 | 1.12 |
| 14 | GI | | 2.3 (0;4) | 0.02 | 0.50 |
| 15 | MC | | 1.51 (0;4) | 0.07 | 0.38 |

The minimum and maximum sizes of fragments produced are also listed in Table 3. Fragments during graupel-graupel collisions have a minimum size of 0.02 mm and a maximum size of 0.24 mm, whereas graupel–ice sphere collisions produced fragments with 0.02 mm minimum and 1.12 mm maximum sizes. In the case of collisions between ice spheres, there were not more than 3 fragments on an average (see Table 3), with fragments having a maximum average size of 1.2 mm and a minimum average size of 0.27 mm. For these ice sphere—ice sphere collisions there seems to be a slight tendency of particle size on CKE. The minimum sizes are comparable to the ice crystal maximum dimensions in G2023, but there are very large fragments within the maximum observed sizes for these collisions. This can most probably be attributed due to the high CKEs.

## 4.2 Compilation of the current results with G2023

Figure 3 depicts a composite plot of the current results together with the bare graupel – bare graupel collisions in G2023. For the G2023 plot, all collisions are taken into account, while for the current results an averaging over the collisions for each CKE are shown. The dashed lines represent the best fits using Eq. 2. For the sake of consistency, we employed the same shape parameter $\gamma = 0.78$ for the G2023 bare graupel data points as for the current graupel – graupel and ice sphere – ice sphere data. G2023 found the best fit on the data if the shape parameter was 0.78 for graupel-snowflake and graupel – graupel with dendrites collisions, while for bare graupel-bare graupel collisions $\gamma = 0.55$ was revealed as the best fit. Nevertheless, when we set $\gamma$ to

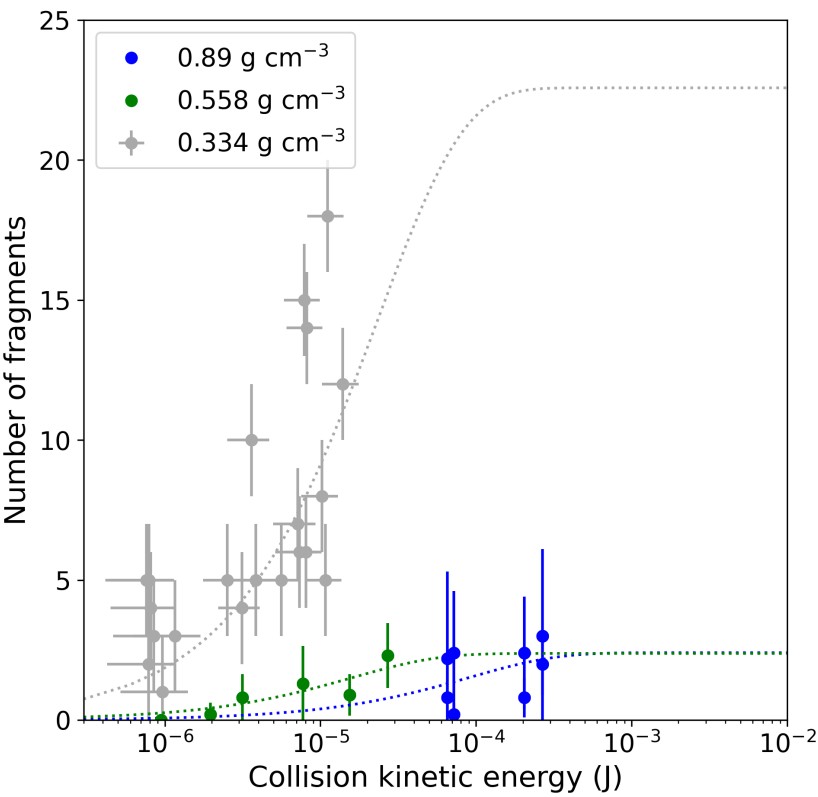

**Figure 3.** Number of fragments generated in bare graupel — bare graupel collisions. Blue: current study, ice sphere — ice sphere collisions (-5 and -15 °C); green: current study, graupel-graupel collisions (-7 °C); grey: G2023 bare graupel– bare graupel collisions (-15 °C). Dashed lines represent the best fits using Eq. 2.

0.78, it modified somewhat the curve but this change was insignificant and well within the measurement error. From Fig. 3, it is obvious that the average fragment number as a function of CKE is dependent on the particle density. The fragment number tends to decrease as the particle density increases. This can be attributed to the fact that the particle's structural integrity and surface properties change with the temperature at which they were generated. The higher density graupels generated at -7 °C are structurally more integrated (see Fig. 1a) than the ones produced at -15°C (Fig. 1b), thus reducing the chance of breaking and generating less fragments during collision. Certainly, ice spheres produce the least number of fragments due to their more compact structure.

The change in particle morphology can also be expressed in terms of the surface density $A$ and fragility $C$ of breakable asperities on the particle's surface (cf. Eq. 2). The surface property of the particles mainly determines the available number of breakable asperities on the surface. If a graupel has a lower density, i.e. more air bubble inclusions are formed when the

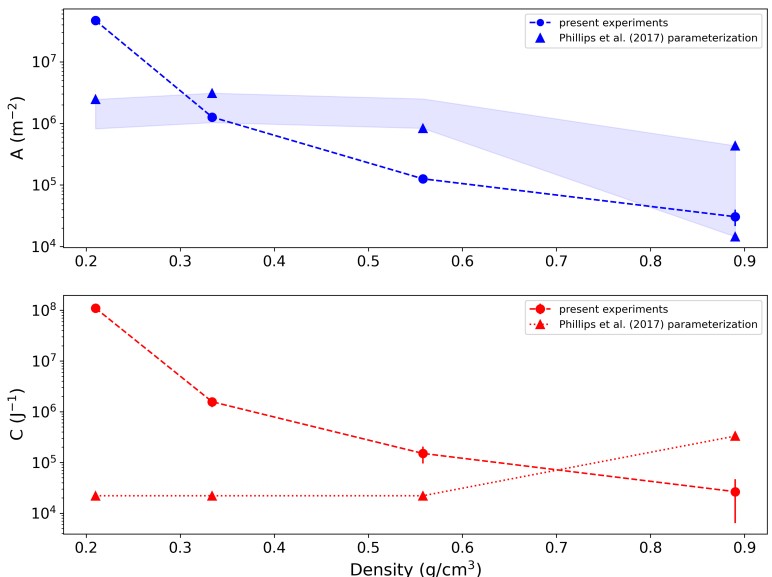

**Figure 4.** Experimentally determined surface density of breakable asperities $A$ (upper panel, blue dots) and asperity-fragility coefficient $C$ (lower panel, red dots) as a function of particle density. Blue and red triangles depict $A$ and $C$ using the Phillips et al. (2017) parameterization. The shaded area is corresponding to a temperature variation from -5 °C to -15 °C using the Phillips et al. (2017) parameterization.

| Density (g/cm$^3$) | $A$ (m$^{-2}$) | $\Delta A$ (m$^{-2}$) | $C$ (J$^{-1}$) | $\Delta C$ (J$^{-1}$) |
|---|---|---|---|---|
| 0.21 | $4.68 \cdot 10^7$ | $1.49 \cdot 10^6$ | $1.10 \cdot 10^8$ | $1.06 \cdot 10^7$ |
| 0.334 | $1.27 \cdot 10^6$ | $1.61 \cdot 10^5$ | $1.57 \cdot 10^6$ | 228785 |
| 0.558 | 126168 | 19953 | 150597 | 55629 |
| 0.89 | 30610 | 9093 | 26534 | 20171 |

**Table 4.** Number density of breakable asperities, $A$, and the asperity-fragility coefficient, $C$, together with their errors from fits on our experimental data in Fig. 3 and G2023 for different particle densities.

particle was generated (Enzmann et al., 2011), it has more asperities than a higher density one. This is obvious also from Fig. 1; graupel generated at -15 °C has more roughness (crevices) on the surface and hence a greater number of breakable asperities than the relatively rather smooth graupel generated at -7 °C. Since the structure of a low-density particle is more loose and fluffy, those asperities are also more fragile, i.e. require less energy to break them off. Consequently, both parameters, $A$ and $C$ should increase by decreasing graupel density. This is reflected in Fig. 4 which shows the variation in $A$ and $C$ as a function of density calculated from the fit curves in Fig. 3. The lowest density particle in Fig. 4 is the graupel with dendrite in G2023, while the one with the highest density is the ice sphere. The parameters $A$ and $C$ together with their errors are given in Table 4.

In their theoretical framework, P2017 proposed the parameters $A$ and $C$ fitted to the experimental data of Takahashi et al. (1995). Following Table 1 of P2017 for Type I collisions, one can calculate $A = 6.8 \cdot 10^6$ to $18.6 \cdot 10^6$ m$^{-2}$ depending on graupel size and temperature, as well as $C = 22050$ J$^{-1}$ and $C = 331000$ for graupel-graupel and ice sphere – ice sphere collisions, respectively. In the P2017 parameterization, both $A$ and $C$ are independent of the particle density. In contrast to these findings, we could demonstrate that $A$ and $C$ are dependent on the density of the colliding particles, thereby we refined and improved upon the previous parameterization. This finding is in accord with the study of Gautam et al. (2024), who presented an improved parameter set for graupel–snowflake collisions.

One has to note here, that we did not carry out any systematic investigation upon the temperature dependency of the fragment numbers. The experiments of Takahashi et al. (1995) indicated a strong variation with temperature, which was also taken into account for in the parameterization of P2017. In agreement with Takahashi et al. (1995) and with P2017, in our opinion this temperature dependency can be attributed to the temperature dependency of the crystal habit, surface density, and size grown on the graupel's surface. For bare graupel and ice spheres such a strong temperature dependency is not expected, but some structural or morphological change is very likely.

The sizes of the graupels corresponding to the three lowest densities investigated in our study were approximately the same. Thus, the largest variation in the generated number of fragments, or in the parameters $A$ and $C$ stems from the temperature variation. In the P2017 parameterization, $C$ is invariant on the temperature. Values of $A$ were calculated for the particle sizes and temperature range of our experiments and indicated by the shaded area in Fig. 4. Since this variation is far less than the observed change in $A$ over the different particle types, one can attribute the systematic decrease of $A$ indeed to the density of the particles.

In Fig. 4, $A$ for ice spheres owns the lowest value both in our experiments and in the P2017 parameterization, having very good agreement at -5 °C. Nevertheless, the predicted temperature dependency of P2017 could not be proven in our measurements. As can be seen in Fig. 2a, the temperature dependency, if any, is within our experimental errors. As mentioned above, the temperature dependency in the P2017 parameterization might be attributed to the temperature dependency of the crystal growth on the ice sphere's surface. Since in our experiments solely the internal structure of the ice spheres can vary due to any temperature change, the resulted temperature dependency becomes minor.

The fragility coefficient $C$ shows similar dependency on the particle density as $A$ in our experiments, but behaves very differently to that of P2017 when employing their Type I parameterization corresponding to graupel-graupel or hail-hail collisions. Nevertheless, the P2017 parameterization was based on a very limited dataset available at that time which prevented the resolution of the full dependencies of $C$ on temperature, size, and rimed fraction. Our experimental data shed light on the dependency of $C$ at least on particle structure (i.e. rimed fraction) and probably temperature.

### 4.3 Multiple collision

In the multiple collision experiments, 2.45-mm-diameter graupels were repeatedly hit by 5-mm-diameter ice spheres until no more fragments were generated on their subsequent collisions. In these experiments, it took two to four collisions to cease generating any additional fragments. On an average 1.5 fragments, and a maximum of 4 fragments were released during these

series of multiple collisions, which match the fragment numbers of single collisions (see Table 3). The number of released fragments was the highest for the very first collision in each set of measurements. The subsequent collisions did not show any

clear tendency for decreasing or increasing number of fragments. Thus, apparently the majority of the breakable asperities broke at the very first collision, and none of them remained after the forth collision. The maximum fragment dimension was 0.384 mm and the minimum 0.072 mm, that also match the values for single collisions (Table 3). Hence, if a graupel undergoes multiple collisions, it behaves like in single-collision events but it stops generating additional fragments after a few collisions. Nevertheless, if the graupel is in the mixed phase zone of a cloud, i.e. it can continuously grow by riming, its structure is

retained in the sense that breakable asperities are continuously produced. Considering the relatively long time between two collision when compared to the growth rate by riming, single collision of graupels can be taken into account when considering secondary ice production.

## 5   Limitations of the present experiments

The main goal of the experiments was to gather more measurement data to improve the understanding and characterization

of secondary ice production after collision of ice particles. The focus was on the effect of the morphological structure of the colliding particles on the number of fragments generated, by exploring the most compact structures of spherical graupel and frozen droplets (ice spheres). We intended to simulate atmospheric ice particles in terms of size, morphology, fragility, and collision kinetic energy. Certainly, this experimental study also has some constraints and limitations. They are listed and discussed in the following, and can be considered in modeling studies and in future laboratory experiments.

– The graupels were generated using epoxy spheres as embryos. In the vicinity of the site where the embryo was connected to the double gyration apparatus, no riming occurred. Therefore, the center of mass of the particle was offset from its geometric center. Due to this eccentricity, the particle was falling with the embryo facing down which would have affected the number of fragments generated during collision. Therefore, the embryos were removed from the falling graupels. After removing the embryo, a hollow space remained inside the graupel which modified its density (from

0.558 g cm$^{-3}$ to 0.46 g cm$^{-3}$). Since the collision occurred at the rimed side of the fixed graupel, such a removal of the embryo from that graupel was not necessary.

– We examined central collisions between the particles, but we cannot rule out any oblique collisions. In G2023 we found that oblique collisions between graupel and snowflakes result in significantly less fragments than central collisions (around 50 % decrease), because some amount of energy was fed into the rotation of the snowflake and not into

fragmentation. For graupel and ice sphere collisions we don't expect such large differences in fragment numbers, because in this case the torque is less than for snowflakes, and, consequently, less energy would be fed into the rotation of the particle. Furthermore, the total number of fragments observed is also of an order of magnitudes less than in case of the more fragile snowflakes.

- In the present experiments, CKEs represent the highest end of the natural collision kinetic energies, and the high particle densities reflects the most compact ice structures. This limits the direct applicability of our results in model simulations. Nevertheless, the main objective of the study was to reveal a relationship of the number of fragments generated due to collision on CKE and on particle density (structure). In the experiments we could capture the lower CKE limits for producing fragments (e.g., close to $10^{-6}$ J for 0.558 g cm$^{-3}$ density graupels). Similarly, we identified the upper CKE limits above which no extra fragments have been generated. Hence, we could cover the CKE range in which fragmentation due to collisions of the investigated particles in the atmosphere could occur. The same holds for particle densities, which represent a wide range of atmospheric ice particle structures, and allows the derivation of a dependency of the number of fragments on this feature.

- The detection limit of our measurements was approximately 25 to 30 $\mu$m. Hence, it is possible that some small ice fragments remained undetected. However, the previous measurement of G2023 showed that the fragment size distribution peaks at 75 and 400 $\mu$m, i.e. well above the detection limit. Certainly, whether small-size ice particles less then 20 $\mu$m survive in the atmosphere, depends strongly on the humidity in the environment. In general, most experimental studies suffer from the relatively poor spatial resolution of the applied optical detection (Lauber et al., 2018; Keinert et al., 2020) and the challenge of distinguishing small ice fragments from dust particles (G2023). Therefore, any comparison of the fragment number derived in the current study with earlier freezing fragmentation (Lauber et al., 2018; Keinert et al., 2020), or collision induced fragmentation measurements (Takahashi, 1993; Vardiman, 1978) should be interpreted with caution.

## 6    Conclusions

Building upon the collision studies by G2023, we investigated the collision between graupel particles of different densities, specifically in the absence of fragile ice crystal growth on their surfaces. Collisions involving graupels grown at different temperatures by riming under dry growth conditions, and ice spheres as proxies frozen drops generated as frozen water were investigated. Combining the dataset of the present study with those from G2023, in which the graupels had densities between approximately 0.2 and 0.34 g/cm$^3$, revealed a direct dependency of the number of fragments generated by collision on the particle's density. This effect can be attributed to the morphological structure of the particles, which ultimately determines both their fragility and their density. P2017 noted that the number of fragments generated increases with the degree of rime of graupels. This increase is explicable in terms of the number density of breakable branches, $A$, as also our study suggests. Our findings contradict the experimental study of Griggs and Choularton (1986) who suggested that fragmentation of rime is very unlikely to occur in natural clouds due to the high energy required to break the structure. Nevertheless, they pointed out that evaporation of rime in an unsaturated (with respect to ice) environment of the cloud would weaken the structure making the rimed graupel more fragile.

P2017 provided a temperature dependent parameterization of the number of fragments produced after collision. We haven't observed as high temperature dependency as P2017, however, the graupels' structure is determined by the temperature at

which the growth occurs. Hence, one can argue that from this point of view our results also revealed an indirect temperature dependency. This suggests that the internal structural integrity of the particle varies with temperature, thereby altering its susceptibility to fracture. Nevertheless, the temperature dependency in P2017 might also be attributed to the temperature dependency of the crystal growth on the ice particle surface in the experiment of Takahashi et al. (1995).

The observed number of fragments produced by graupel- graupel collisions is within the range or higher than those observed in other secondary ice processes during drop freezing Lauber et al. (2018); Keinert et al. (2020), at least for the sizes detectable by the applied optical imaging techniques. However, as showed in G2023 for graupel-snowflake and graupel-graupel with dendrites collisions, when ice particles possess vapor grown fragile ice structures, the number of ice fragments can rise up to hundred for one collision. The typical cloud regime range for collision induced fragmentation of frozen drops might coincide to that for fragmentation of drops during freezing (Lachapelle and Thériault, 2022), whereas fragmentation of graupels would occur at lower cloud regimes and at temperatures between -5 °C and -20° C. This regime coincides with the typical H-M range of -3 to -8 °C (Hallett and Mossop, 1974), however, the observed number of fragments is far less than those suggested from H-M processes. Therefore, we suggest sensitivity studies using cloud models to explore the role of collision induced fragmentation of graupels in the H-M regime in ice multiplication and cloud and precipitation development. Although numerous studies have highlighted the important role of collision induced fragmentation of ice particles in convective (Waman et al., 2022; Han et al., 2024; Grzegorczyk et al., 2024), Arctic (Sotiropoulou et al., 2020, 2021; Karalis et al., 2022), and orographic mixed-phase clouds (Dedekind et al., 2021; Georgakaki et al., 2022), further laboratory investigations are needed to quantify this process under varying atmospheric conditions that influence particle properties.

*Data availability.* The data used for generating the figures are provided in the tables of the paper or in Grzegorczyk et al. (2023). The raw measurement data will be provided upon request.

*Author contributions.* The paper was written by M.S. and S. Y. with the support and assistance of all co-authors; P.G. made significant contributions by providing comments on the results, discussion, and conclusion, as well as in data evaluation; S.Y. performed graupel growth and collision experiments, and evaluated the data; L.M. performed ice sphere collision experiments and analyzed the data; A.T. constructed the graupel generator, designed the graupel growth experiments; S.K.M. and S.Y. designed the graupel growth and collision experiments and characterized the setups; M.S. designed the experiments, analyzed the data.

*Competing interests.* The authors declare no competing interest.

*Acknowledgements.* This work was funded by the Deutsche Forschungsgemeinschaft (DFG, German Research Foundation) – project number 492234709. We gratefully acknowledge the funding of the German Research Foundation (DFG) to initialize the special priority pro-

320 gramme on the Fusion of Radar Polarimetry and Atmospheric Modelling (SPP-2115, PROM). We also gratefully acknowledge the help of Dr. Christoph Sievert (German Weather Service) in the calculation of the collision kinetic energy of atmospheric graupels and ice particles. Pierre Grzegorczyk is now funded by the French National Research Agency (ANR) (ACME ANR-21-CE01-0003 project contract).

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
