# Peer review of "Measurement Report: Influence of particle density on secondary ice production by graupel and frozen drop collisions"

_EGUsphere, 2024_

## Author Response (AR1)

We greatly appreciate the reviewers' careful reading and review of this manuscript. We have addressed each reviewer's comments and suggestions, as shown in our responses below.

We added several new references to the literature list, which might not be highlighted in the track changes version of the manuscript.

The reviewers' comments are written in bold, the authors' replies in normal fonts.

**Authors' response to RC1 – Referee#1**

*General comments:*

*In their experimental study Yadav et al. quantify the number of fragments generated upon collisions between ice pellet and graupel particles and their dependency on particle density. This study extends the work of Grzegorczyk et al. (2023), enhancing our understanding of the collisional break-up mechanism—a critical process in mixed-phase cloud microphysics that is increasingly supported by observational and modeling evidence. Current parameterizations of this process rely on limited and outdated laboratory data, underscoring the importance of this study in providing updated experimental insights to improve its representation in cloud models. Prior to publication, the following major comments should be addressed, followed by minor suggestions and editorial remarks:*

**Specific comments:**

**• In the Experimental section, the authors should discuss the extent to which the ice particles used in the study reflect natural atmospheric conditions. For example: Under what synoptic conditions and cloud configurations would graupel and ice pellets of the sizes used (2.45 mm for graupel; 5 mm and 7 mm for ice pellets) form? Is the liquid water content (LWC) used in the experiments realistic for these scenarios? Additionally, the prevailing supersaturation with respect to ice is not mentioned, yet this parameter is important for ensuring a more accurate implementation of the results in cloud models.**

Hail embryos located in the dendritic growth temperatures can reach sizes up to 20 mm (Adams-Selin and Ziegler, 10.1175/MWR-D-16-0027.1 and the references therein). This occurs of course in the high updrafts of hail producing storms, but it implies that graupels and frozen drops of 2 to 5 mm sizes might frequently be produced in convective mixed-phase clouds. Furthermore, Takahashi (1993) observed during his aircraft measurement flights graupels with sizes of 2 to 4 mm in a convective cloud. In such clouds, the LWC is very similar to that in our experiments. Following the suggestion of Reviewer#3, we added a table of the conditions used for graupel generation, which also includes the supersaturation with respect to ice.

**• The study contains several limitations that should be discussed more thoroughly in the results and conclusions sections:**

**- First, the experiments focus solely on central collisions, unlike Grzegorczyk et al. (2023), which also examined edge collisions. The potential impact of this choice on the number of fragments generated (e.g., whether head-on collisions are expected to produce more fragments than oblique ones), should be addressed.**

In the experiments we did not differentiate between edge and central collisions, although that has impact on the number of fragments generated. Nevertheless, we do not expect any high impact due to the anyway small number of fragments generated (and detected). In the case of graupel-snowflake collisions (like in Grzegorczyk et al. 2023), the impact is expected to be higher, because the snowflake is larger and less spherical. That results in a large torque when in case of edge collision, and a relatively large part of CKE is fed into rotational energy. The torque for a graupel is expected to be less, and therefore, more energy is fed into fragmentation.

**- Second, the authors acknowledge that the collisional kinetic energies (CKEs) used in their experiments represent the upper range of natural CKE values (Lines 124-129), which could influence the applicability of the results to natural conditions.**

In the experiments we could capture the lower CKE limits for producing fragments. Similarly, we identified the upper CKE limits above which no extra fragments have been generated. Hence, we could cover the CKE range in which fragmentation due to collisions of the investigated particles in the atmosphere could occur.

**- Finally, the detection limit for identifying small fragments (20 μm; Lines 137-140) imposes a constraint, particularly on the observed size of fragments in graupel collisions mentioned in Lines 159-160.**

We agree on that. Due to the comments of both reviewers regarding the detection limit, in the revised manuscript this issue is more thoroughly discussed. Here we just note, that already for collisions including graupels with dendrites on their surface, we could detect a mode in the size distribution at 75 micron, i.e. above our detection limit. That implies that the detection limit is appropriate for studying the fragmentation of graupel and frozen drops, as well. Certainly, we cannot rule out that there were no small-size ice fragments generated by the collision.

**These limitations should be explicitly acknowledged and discussed in both the results and conclusions sections to provide a balanced interpretation of your findings.**

We agree with the reviewer, and decided to present a stand-alone session for discussing the limitations of the experiments. This is Session 5 of the revised manuscript.

**• The statements made in Lines 192-194 and 216-217 are rather bold and should be tempered by a discussion of the experimental limitations. Specifically, the limited number of experiments conducted, the use of artificial graupel and ice pellets, which may not fully represent natural particles, and the need to examine a broader range of ice particle sizes to cover the phase space relevant for numerical cloud models. Recent work by Gautam et al. (2024) used field observations to constrain the empirical parameters in the theoretical framework developed by Phillips et al. (2017). Comparing the experimental results in this study against these new findings would strengthen the discussion and enhance the robustness of the conclusions.**

Because of the limitations of the experimental study we wanted to avoid hypothetical formulation that could not be supported by our results. We could present the dependency of the parameters A and C on particle density, which is the main result of the present study. Since Gautam et al. (2024) investigated graupel-snowflake collisions, and not graupel-graupel collisions, no direct comparison with their result was possible. Nevertheless, both studies showed that particle morphology is crucially determines the fragility during collision.

**• Both the abstract and conclusion sections should better articulate the motivation (the importance of ice multiplication and the understudied contribution of ice pellets within this context), as well as the broader implications of your study.**

We modified the abstract and the conclusion following the reviewer's suggestion to emphasize the important and yet understudied role of collision induced fragmentation. We also added a sentence at the end of Conclusion to place our study across many modeling studies showing the importance of this process, which therefore highlight the relevance and the motivation of studying the SIP involving graupel and ice pellets.

**Minor comments:**

**• Line 11 and Line 226: Consider replacing "suffers" with "undergoes" or "experiences" to improve phrasing.**

Thank you, we modified the text accordingly.

**• Line 15: Maybe use "formed" instead of "generated" here, to avoid repetition.**

We modified the text accordingly.

**• Lines 17-19: Specify the expected temperature range within the mixed-phase regime where SIP is anticipated to address discrepancies between ice crystal number and ice nucleating particle concentrations.**

In the Experimental section of the revised manuscript we present the experimental conditions in Table 1.

**• Line 20: Here we could acknowledge not only the lack of systematic laboratory studies but also the challenges in identifying SIP in field observations and**

**accurately representing these small-scale cloud processes in model grid cells across different horizontal resolutions (through empirical parameterizations).**

We rewritten the motivation of study, and focus mainly on the structural and morphological dependence of ice particles. We intended to highlight the challenges in identifying SIP in field studies in Introduction, and in modelling studies in Conclusion.

**• Line 22: Note that both droplet diameters <13 μm and >24 μm should coexist in Hallett and Mossop (1974).**

Thank you, we provided the more accurate conditions.

**• Line 52: Are you referring to a specific publication here about the importance of SIP in thunderstorm clouds?**

The leak of publications and the vague formulation of this part of the Introduction was pointed out by both reviewers. We therefore rewritten the motivation of study, and focus mainly on the structural and morphological dependence of ice particles.

**• Line 55: Please clarify "underestimated" environment. Did you mean "undersaturated"?**

Yes, thank you for the comment, we corrected the phrase in the revised manuscript.

**• Lines 55-59: Please include references to support this part of the introduction. Are you referring to observational studies (e.g., Korolev et al., 2022; Lachapelle et al., 2024; Lachapelle and Thériault, 2022) or laboratory-based research?**

We rewritten the motivation of study, and focus mainly on the structural and morphological dependence of ice particles.

**• Line 62: Could you clarify what you mean by "fragmentation outcome" here? The number and/or size distribution of the particles produced?**

Following the second reviewer's suggestion, we modified the sentence and deleted "outcome".

**• Line 70: Maybe "lower" instead of "less"**

Modified, thank you.

**• Line 72: Please define what GEORG stands for here.**

The definition of the acronym is given now in the revised manuscript.

**• Figure 1 caption: Please provide additional details for the ice pellet shown in panel c, including its density and the temperature at which it was formed, to align with the information given for graupel particles in panels a and b.**

Done, thank you for the comment.

• **Table 1: For GG collisions (Experiments 9-11), please confirm whether the density of 0.46 g/cm$^3$ for graupel particles is accurate. It does not appear in Figure 4, Table 3, or the main text. Could this be a typo?**

The graupels were formed by riming, for which epoxy spheres was used as embryos. In several cases, the rime did not completely cover the embryo, which offset the center of mass from the geometric center of the graupel. This offset resulted in an unnatural falling mode, in which the graupel collided with its epoxy embryo facing to the stagnant graupel. To avoid this unnatural collision, we removed the epoxy sphere from the falling graupels before the experiments, therefore their density was 0.46 g/cm$^3$. This issue is discussed in lines 119 to 122 of the revised manuscript.

• **Table 3: Please confirm whether results for a density of 0.21 g/cm$^3$ from Grzegorczyk et al. (2023) are also included in Figure 3.**

No, here we plot only the data of bare graupel – bare graupel collisions.

• **Figure 4: Consider adding legends to enhance readability. Additionally, include the temperature at which the experiments were conducted alongside the density information. This will help readers more easily identify the points referenced, such as those mentioned in Lines 207-208.**

We added legends to the plot. We decided to not include the temperature in the plot because in our opinion it made the figure to busy.

• **Line 112: Line 112: Have you specified these predefined conditions used for generating graupel particles somewhere in the text?**

We provided these conditions in the revised manuscript.

• **Line 252: "possess"**

Corrected, thank you.

• **Line 256: Maybe you could reiterate the Hallett-Mossop temperature range here to avoid potential confusion with the wider -20°C < T < -5°C range mentioned earlier.**

Done, thank you for the comment.

**Authors' response to RC2 – Referee#3**

*In the measurement report results from collisions of different larger ice particles agents such as graupel and ice pellets, at different temperature and ice particle density or surface roughness are summarized and discussed. The Phillips (2017)[1] approach was applied to relate the number of secondary ice fragments to the kinetic energy of the collision. In addition, the results were compared to a previous study where bare graupel-bare graupel collisions were investigated [2]. The topic is relevant, interesting and fits within the scope of Atmospheric Chemistry and Physics. However, the manuscript needs revision in a few points before I recommend publication.*

*The measurement report is comprehensive and discusses some aspects very well, but it lacks precision in its formulation and explanation. Sometimes some unimportant aspects are described in too much detail. The main outcome can be presented more clearly in the abstract.*

**One general question is: How often do graupel-graupel or even graupel-ice pellet collisions take place in a real cloud or in modelled clouds? How likely is that? I strongly suggest, that the authors give a short elaboration on that in the introduction.**

The collision rate of graupels might indeed be very low. Considering a typical graupel concentration in mixed-phase clouds of 5 per $m^3$, 2 to 3 m/s fall speed, and a residence time of 1000 seconds, 1 collisions per graupel can be calculated. Whether the fragmentation is important at such low collision rates should be studied by numerical cloud models. Nevertheless, the graupel concentration might be larger in convective clouds. Unfortunately, remote sensing instruments, like radars cannot resolve microphysical processes, and there are not too many in situ observations in this regard due to obvious reasons. Nonetheless, Takahashi (1993) carried out aircraft measurements in maritime convective clouds and he observed high graupel observations, and hypothesized that graupel-graupel collisions might be the reason of the high concentration of ice crystals observed.

We rewritten the motivation of study, in order to mainly focus on the structural and morphological dependence of ice particles, and not on the direct effect of graupel-graupel collisions on the secondary ice production in clouds.

**P1 l9: The number of ice fragments produced by drop freezing experiments is not clear yet as it is so far based only on detection with high speed microscopy [3, 4]. In Kleinheins et al. (2021) [5] indications were found for pressure-release events likely producing secondary ice splinters, which could not be seen high speed microscopy detection. A more detailed discussion on the detection limits and value of the statement is required when such a comparison is made.**

We agree with the reviewer, and decided to present a stand-alone session for discussing the limitations of the experiments. This is Session 5 of the revised manuscript.

**P5 l138: The detection limit has to be mentioned already in the abstract.**

Done, thank you for the suggestion.

**P12 l250-251: Again. This cannot be states, see comments above.**

We tried to clarify the issue of the detection limit throughout the whole manuscript, also here.

**P2 l27-28: It is important to note and present properly in the report, that the number of secondary ice particles could only be detected visually and might have therefore a large bias towards larger ice splinters. The quantification of smaller ice particles is not possible at the moment. Care should therefore be taken to ensure that no false impression is created.**

The issue is now addressed in the abstract, but also the new Session 5 aims to emphasize this.

**P3 l59: Why is this temperature range explicitly given? Why is it not larger and goes up to 0°C? I would expect the temperature to be of minor relevance and thus only the existence of these two colliding objects and their kinetic energy might be of major importance.**

We agree with the reviewer, that the temperature could go to 0 °C. The upper limit of -3 °C was given to emphasize the coincidence with the H-M-range. We have rewritten this part of the Introduction, and deleted this sentence.

**Minor comments:**

**P1 l1/l4: The difference of 'graupel' and 'ice pellet' has to be clarified already in the abstract. Are ice pellets referred to freezing rain? This is necessary to understand the concept of these measurement reported.**

Thank you for this comment. We used the phrase "ice pellet" for "frozen drop". We corrected for this issue throughout the manuscript, and used only "frozen drop". Also the title of the paper has been changed accordingly.

**P1 l9-10: Information of the detection limit is needed. A real comparison is only possible if the detection limits are clear and comparable and if no smaller secondary ice particles would exist, which is not clear. The detection should be mentioned and the statement should be changed or removed (see major concern).**

Done, thank you again for pointing out the issue with the detection limit.

**P2 l50-52: Could you confirm this statement with a reference?**

The leak of publications and the vague formulation of this part of the Introduction was pointed out by both reviewers. We therefore rewritten the motivation of study, and focus mainly on the structural and morphological dependence of ice particles.

**P2 l52-54: This sentence is confusion. Which agents are colliding with each other? What is the relation to the abundance? Thunderstorms are also mixed-phase clouds. Maybe you can use the term of 'convective' for differentiation.**

This part of the Introduction has been rewritten, and we mainly focus now on the morphological and structural dependence of the particles on the fragmentation.

**P2 l54: What is meant by strength? Please clarify.**

We used the same formulation as Griggs and Choularton in their study. The strength actually indicates how non-fragile the rime is.

**P3 l55: What is meant by 'underestimated environment'? Subsaturated with respect to ice? Clarify.**

Thank you, corrected.

**P3 l80: It is unclear what 'unwanted size or density' means. Shorten the sentence and avoid unnecessary information.**

Thank you for the comment, that was a bad formulation. We rewritten the text, which now reads as "a relatively large scatter in graupel size and density during the preparation phase of our experiments".

**P3+ section 2.1: A table presenting all these multiple values would be much clearer and easier to understand.**

We added Table 1 with the experimental conditions into the revised manuscript, and modified the text accordingly.

**P4 l104-105: I think that this information is not needed.**

We agree and deleted the sentence.

**P5 l111-112: I don't think that this information is understandable in its actual form 'during the characterization measurements after producing several particles in GEORG under predefined conditions, like temperature, growth time, LWC.'**

We added the parameters in order to clarify the conditions.

**P5 l113-116: Is this a relevant information? How does it affect the collision as the mass distribution is different? In case, a discussion would be needed.**

We think this information is relevant. During the test experiments we observed that the epoxy inside a falling graupel modified the center of mass of the particle. This resulted in a fall orientation in which the unrimed epoxy hit the fixed graupel. In order to avoid this

unnatural collision mode, we removed the epoxy from the falling graupel particle before the collision experiments. This discussion in now added to the text, and the issue is also addressed in Session 5.

**P5 l117: Why is it random?**

We deleted the word random, which was indeed false in this context.

**P5 l 121: I strongly recommend to rephrase this part to : 'no new ice fragments could be observed considering the detection limit of ...?'**

We reformulated this sentence for clarity.

**P5 l122: I don't understand why a new fragile graupel does not produce ice fragments after collision.**

In this sentence, we mention graupel instead of frozen in which was used to repeat multiple collisions. The sentence which is now corrected.

**P5130: Does the oil conserve the shape of ice particles, so that a size distribution can be derived for larger fragments? Or do they melt and only equivalent droplet diameters can be derived from that?**

The shape of the ice particle was conserved, since the oil was placed inside in the clod chamber, i.e. cooled to the cold chamber temperature. Please see Figure 8 and corresponding description in Grzegorczyk et al. (2023).

**P5 l134-136: I don't think that this is the reason. Maybe the fragments were too small? Or the shape was not conserved? Or I could simply leave the last part of the sentence out.**

We reformulated the sentence and pointed out the issue with the detection limit. It is indeed possible that there were more fragments which we could not detect, but we do not expect very high number because of the compact structure of the particles investigated. As mentioned in the paper, because of the small number of fragments observed, it was not possible to calculate a size distribution.

**P5 l139-140: I would delete this sentence. It is confusing and give no further insights.**

We deleted this sentence.

**P8 l178: What is meant by 'bulk structure'. Do you mean a 'more compact structure'?**

Yes, thank you for this comment, we modified the text accordingly.

**P12 l240-243: That is not completely true. The information of the frequency of graupel-graupel collisions in atmospheric clouds is missing. Please clarify and**

**elaborate a bit more on this. Do you mean 'unsaturated' with respect to ice? What is the consequence of this argument (242-243)?**

If the environment is unsaturated with respect to ice, sublimation would weaken the structure of the ice particle, therefore the rime would be more fragile. We modified the sentence accordingly.

**P12 l255-256: Specify the reference to which you refer. H-M experimental results are very different and inconsistent [6, 7].**

We added the reference to specify the results.

**Technical corrections:**

**P3 l55-56: You can include an 'also' as this sentence is not directly related to the previous sentence, or reformulate.**

**P3 l62: Remove 'outcome'.**

**P4: Change 'into' to 'of'? and delete 'any'.**

**P4 l100: I am not sure if 'kept stagnant' is a proper formulation.**

**P8 l175: I would delete 'change in'.**

**P10-11 l204-205: I would delete the sentence in brackets.**

**P11 l233: Do mean 'consequences' instead of 'outcomes'?**

We appreciate the reviewer's detailed suggestions for technical corrections listed above. We have carefully considered all of them and revised the text accordingly.

---

## Author Response (AR2)

We thank **Referee#1** for reviewing also the revised manuscript. We corrected all the mistakes and confusing formulation listed by the Referee.

We would like to thank **Referee#4** for reviewing our manuscript. In the following we reply to the Referee's comments.

**• Line 9 (Abstract): Suggest clarifying "The lower detection limit..." and correcting the formatting to "30 μm".**

For the sake of clarity, we revised the sentence to only mention the upper end of the detection limit (30 μm), which sufficiently represents the sensitivity of our setup. We also corrected the unit.

**• Line 10 (Abstract): For better flow, consider moving the sentence "Our results revealed a strong dependency..." to follow the sentence "The observed number of fragments varies...".**

Thank you for the suggestion, we modified the abstract accordingly.

**• Line 21: To clarify the discrepancy, explicitly state that observations show "more ice crystals than ice nucleating particles (INPs)". Table 1: The RH range is given as 92-100%. Could the authors explain the reason for the range? Was it measured or controlled? Slight variations in RH could potentially influence surface properties or sublimation effects. Remove double dash between 92 and 100.**

The observations showing this discrepancy are presented or cited in the publications given at the end of the sentence. The Ladion 2017 study, for instance is an observational study showing the discrepancy between INP and ice crystal concentrations. The GEORG facility utilizes a continuous stream of supercooled droplets carried with an air flow at around 3 m/s. We measured the dew point at the place where the graupels are growing. Because of the fluctuation caused by the inhomogeneous distribution of droplets, and the slightly fluctuating temperature, we could only estimate the relative humidity with respect to ice from the measured dew point values. This is the reason why a range for RH is provided. The double dash has been removed.

**• Lines 100-102: Ice spheres were generated at $-70°C$ and then equilibrated at experiment temperatures. Does the initial very low freezing temperature potentially influence the internal structure (e.g., bubble concentration/distribution) compared to spheres frozen directly at $-5°C$ or $-15°C$, possibly affecting fragility?**

Yes, that might be possible. Our aim was to generate ice spheres with a density of bulk ice, and to avoid any fluctuations in the structure. For that, freezing water at low temperature, so thus, also quickly, seemed to be the proper way. This limitation of the experiment is now added to Section 5.

**• Line 118: Correct formatting to add space: "0.46 g cm$-3$".**

Done.

**• Table 2: Consider simplifying the table by showing repeated values (e.g., fixed particle size) only once or using grouping. What is the uncertainty associated with the CKE values? Please reduce CKE values to a maximum of two significant digits for consistency and clarity.**

We chose this representation of the table because the particle pairs are varying. In our opinion, grouping the repeated values would make the table more confusing. The uncertainty of the CKE values was 4.3%, which is now also provided in the text.

• **Table 2 & Table 3: Suggest combining Table 2 (experimental conditions) and Table 3 (fragment results) into a single table. This would allow readers to more easily connect the collision parameters with the resulting fragment numbers and sizes for each experiment series.**

We decided to present the experimental conditions and the results in two distinct tables. Merging the tables would indeed allow a direct view of the collision parameters and the resulting fragment numbers, but it would make the table too big and probably reduce the overall readability.

• **Table 4: For consistency and clarity, please format the uncertainties using the "value ± uncertainty" notation. Also, reduce the number of significant digits for A, C, and their uncertainties to a maximum of two.**

We modified Table 4 following the Referee's suggestion.

• **Lines 181-182: When discussing structural integration, explicitly reference the figures mentioned: "...structurally more integrated (see Fig. 1b) than the ones produced at −15∘C (Fig. 1a)...".**

Corrected. Thank you for finding this.